# Harnessing the Power of Mast Cells in unconventional Immunotherapy Strategies and Vaccine Adjuvants

**DOI:** 10.3390/cells9122713

**Published:** 2020-12-18

**Authors:** Steven Willows, Marianna Kulka

**Affiliations:** 1Nanotechnology Research Centre, National Research Council Canada, 11421 Saskatchewan Dr, Edmonton, AB T6G 2M9, Canada; Steven.Willows@nrc-cnrc.gc.ca; 2Department of Medical Microbiology and Immunology, University of Alberta, Edmonton, AB T6G 2E1, Canada

**Keywords:** mast cells, innate immunity, adaptive immunity, wound healing, immunoglobin E, vaccine adjuvants

## Abstract

Mast cells are long-lived, granular, myeloid-derived leukocytes that have significant protective and repair functions in tissues. Mast cells sense disruptions in the local microenvironment and are first responders to physical, chemical and biological insults. When activated, mast cells release growth factors, proteases, chemotactic proteins and cytokines thereby mobilizing and amplifying the reactions of the innate and adaptive immune system. Mast cells are therefore significant regulators of homeostatic functions and may be essential in microenvironmental changes during pathogen invasion and disease. During infection by helminths, bacteria and viruses, mast cells release antimicrobial factors to facilitate pathogen expulsion and eradication. Mast cell-derived proteases and growth factors protect tissues from insect/snake bites and exposure to ultraviolet radiation. Finally, mast cells release mediators that promote wound healing in the inflammatory, proliferative and remodelling stages. Since mast cells have such a powerful repertoire of functions, targeting mast cells may be an effective new strategy for immunotherapy of disease and design of novel vaccine adjuvants. In this review, we will examine how certain strategies that specifically target and activate mast cells can be used to treat and resolve infections, augment vaccines and heal wounds. Although these strategies may be protective in certain circumstances, mast cells activation may be deleterious if not carefully controlled and any therapeutic strategy using mast cell activators must be carefully explored.

## 1. Intro/Background

In the early 1960s several observations implicated mast cells and immunoglobulin E (IgE) as mediators of atopy, and animal models of passive cutaneous anaphylaxis showed that mast cells regulate allergic inflammation by production of several proinflammatory mediators, including histamine [1,2,3,4]. In the following decades, mast cells were often studied in the context of allergic inflammation. However, in 1991, Margaret Profet proposed the toxin hypothesis, which theorized that allergies evolved through Darwinian natural selection as a defense against toxins, carcinogens and mutagens [5]. Early work by Echtenacher et al. showed that mast cells were protective against infections as well [6]. Though understudied, the postulate that mast cells are protective in some circumstances persisted, and in recent decades mast cells have been established as important contributors to homeostasis and initiation of protective and repair mechanisms in tissues. Mast cells have a sophisticated receptor repertoire, including receptors that recognize pathogen-associated molecular patterns (PAMPs) and damage-associated molecular patterns (DAMPs), which sense pathogens and alarmins respectively, and this allows mast cells to tailor their response to each aberrant circumstance and signal to other cells to initiate a defensive/protective response. In wound healing, these responses coordinate a complex web of angiogenesis, inflammation, adaptive immune cell activation, enzymatic degradation of extracellular matrix and related processes to restore homeostasis. At the same time, in some microenvironments, mast cells contribute and amplify disease pathology. Ultimately, a greater understanding of mast cell responses to specific stimuli and the mast cell-specific responses that are initiated allows for harnessing of these pathways to develop novel and more precise immunotherapeutics.

Mast cells originate from the bone marrow and circulate as immature progenitors in the blood until finally maturing in the tissue in which they reside for the remainder of their life [7]. Some evidence suggests that mast cells instead mostly originate from local progenitor cell pools initially established by successive waves of colonization that occur during embryogenesis [8,9,10,11]. While mast cells derived from earlier waves tend to be replaced by mast cells from later waves, the kinetics of these changes vary by tissue [10]. Notably, mast cells from different waves of colonization exhibit different gene expression profiles, leading to heterogeneity even within a given tissue [9,10]. Stem cell factor and its receptor c-Kit are essential for proper development of mast cells from their progenitors, although many other cytokines influence their development as well [7]. Mast cells are distributed widely throughout the body, especially at surfaces that contact the external environment like the skin and mucous membranes, allowing them to act as one of the first responders to infection, toxins and injury [12]. Mature human mast cells are generally divided into two different subpopulations based on their protease content: those with tryptase (MC_T_) equivalent to mucosal mast cells in mice, and those with both tryptase and chymase (MC_CT_) equivalent to connective tissue mast cells in mice [13]. Mast cell heterogeneity extends beyond this binary division, though, and mast cells can express different receptors, proteases and responses to external stimuli based on the stage of development and signals from their surrounding microenvironment [8]. Though not discussed in this review, it is important to keep in mind that mast cell phenotype can vary significantly, potentially limiting the applicability of treatments discussed below to situations other than those assessed in a given study.

As first responders of the innate immune system, mast cells utilize a plethora of cell surface receptors to sense pathogens. The best known is the high affinity receptor for IgE: FcεRI. This receptor binds strongly to IgE and is in turn activated by crosslinking of IgE to antigens associated with pathogens, venoms or allergens [14]. Basophils, which are similar to mast cells and release histamine filled granules, also express FcεRI and are involved in allergic responses [15]. The low affinity IgE receptor (CD23) is expressed on several cell types of hematopoietic origin, but not on mast cells [15]. Mast cells additionally express IgG receptors, FcγRI and FcγRII in humans and FcγRII and FcγRIII in mice, that can either potentiate or inhibit mast cell functions [16]. Mas related G protein coupled receptor X2 (MRGPRX2) is activated by cationic amphiphilic molecules, including the classical “mast cell activator” compound 48/80 (c48/80), but this receptor is only expressed by a specific subphenotype of mast cells associated with connective tissues [17,18]. Mast cells express many G protein-coupled receptors (GPCR), which bind lipids, proteins and nucleotides and these appear to either activate or inactivate mast cell functions. Nedocromil sodium, cromoglycate, sodium cromoglicate or cromolyn sodium are often used to “inactivate mast cells” and although, their molecular targets are still poorly understood, they may target GPCR 35 or chloride channels on mast cells [19]. Mast cells express several toll like receptors (TLRs), which are activated by PAMPs and cause de novo synthesis of mediators but not degranulation [20,21]. Several other classes of receptors expressed by mast cells can detect a wide range of perturbations in the local cellular environment, including cytokine release, cell damage and even vibrations [18,22]. The long list of receptors expressed by mast cells and the mast cell mediators that are produced by their ligation have been comprehensively reviewed by others and this review will focus primarily on a few that may be significant in therapeutic strategies [18,23,24].

Mast cells are filled with large granules containing preformed signaling molecules and proteases. When activated through their receptors, including the various Fc receptors and MRGPRX2, mast cells quickly release these granules, allowing a rapid response to external stimuli. Many of the classic symptoms of allergic reactions, like urticaria, congestion and a drop in body temperature, are initiated with the release of histamine, which is stored in a proteoglycan matrix in a subpopulation of granules [25,26,27]. Preformed cytokines, most notably TNF-α, recruit leukocytes and help fuel local inflammation [23] while growth factors, like VEGF, TGF-β1, PDGF and FGF, contribute to angiogenesis and wound healing [23]. Several proteases, such as tryptases and chymases (the most abundant), cleave their target proteins to promote or inhibit inflammation, promote coagulation, and reduce the integrity of the extracellular matrix [28]. Mast cells can also have longer term effects through de novo synthesis of various proteins and mast cells are able to “re-arm” by forming new granules and degranulating multiple times [29].

As appreciation for the protective roles of mast cells has grown, many strategies that could result in therapeutics targeting these cells are being tested. Since mast cells are involved in so many biological processes, this review will focus on therapeutic interventions that have a direct effect in vivo, especially when the effect of the therapy in question is reduced in mast cell deficient mouse models. This comes with the caveat that controversy exists as to the applicability of many of these models to human pathology [30], especially in animal models that rely on *c-kit* mutations, since this receptor is expressed in many other cell types in mice [30]. Furthermore, differences exist between the distribution and phenotype of mast cells in animal models, especially mice, and humans. For example, in contrast to humans, mast cells in mice are rare in the lung parenchyma and preferentially secrete 5-hydroxythryptamine rather than histamine when activated by IgE, which affects their validity as an animal model of asthma [31]. Human and mouse mast cells also express different subsets of tryptases and chymases [28]. The mouse orthologue of MRGPRX2, MrgprB2, differs in its sensitivity to various stimuli [32]. Human mast cells also respond differently to cytokines and growth factors as compared to mouse cells and their mouse orthologues, requiring different subsets for proliferation and maturation [8]. Finally, mast cells can also express different receptors on their surface depending on the tissue they are isolated from [8]. Since therapies targeting mast cells in the treatment of atopic disease have been discussed extensively elsewhere [33,34], this review will instead focus on three other key areas where mast cells play a role: innate immunity, adaptive immunity and wound healing.

## 2. Innate Immunity

Innate immunity includes all components of the immune system that are not specific for a single pathogen or set of antigens, but rather rely on non-specific defenses or common molecular patterns associated with groups of pathogens (PAMPs) or damage (DAMPs). These patterns are recognized by receptors that do not vary between different cell “clones” in contrast to receptors of the adaptive immune system. This allows a more rapid response, not requiring time for antigen presentation or clonal expansion, but comes at the cost of specificity and the risk of promoting pathogen resistance. Mast cells are some of the first cells to respond to infection and damage, acting as essential parts of the innate immune system. Aside from their role in allergies, mast cells are probably best known for controlling parasite infections. Through multiple mechanisms, both independent of IgE and through IgE mediated degranulation, mast cells are essential for expulsion of nematode parasites in the intestine [35]. Though mast cells also seem to play a role in controlling single cell, protozoan infections such as those caused by *Plasmodium*, *Trypanosoma*, *Leishmania* and *Toxoplasma* species, whether they are beneficial or harmful remains unclear [36] and mast cells can appear protective or harmful depending on the model system, parasite type and infectious dose. Besides parasites, mast cells and IgE play a major role in protecting from venoms and toxins [14,37] where cell damage or antigen-specific IgE cause mast cell degranulation and preformed proteases cleave the venoms, rendering them inactive. Initially, it was thought that the ability of venoms and toxins to activate mast cells contributed to their pathology, but an early study by Metz et al. showed that mast cell activation was pivotal to reducing venom toxicity [38]. Mast cells can also cleave and deactivate Endothelin-1, a peptide produced by endothelial cells that contributes to pathological vascular changes during sepsis [39]. Interestingly, a snake venom, sarafotoxin 6b and Endothelin-1 share homology and are inactivated by cleavage of 1–2 c-terminal residues by mast cell carboxypeptidase A [40]. Mast cells play an essential role in fighting fungal, bacterial and viral infection. Mast cells are able to recognize many different PAMPs, mainly through the use of cell surface pattern recognition receptors (PRRS) such as TLRs [20,21]. What TLRs are actually expressed on mature mast cells is currently a matter of debate, with different studies showing conflicting results [41]. Surprisingly, a proteome study on human and mouse primary mast cells found an absence of many innate immune sensors, including all TLRs [42]. Despite this, TLR agonists can cause changes in gene expression, release of inflammatory mediators and mast cell migration in several models [43]. While some studies show otherwise, TLR activation is generally not thought to lead to rapid degranulation as seen with activation of FcεRI [41,43]. Mast cells can also utilize Dectin-1 to sense β-glucans from fungi [44] and the intracellular receptors RIG-I, MDA5 and TLR3 to detect viruses [45,46,47]. Several innate immune cationic peptides with antimicrobial activity, such as defensins and cathelicidins, can also cause mast cell chemotaxis and degranulation through activation of MRGPRX2 [48]. These peptides, which are released by neutrophils and epithelial cells, further connect other components of the innate immune response and mast cells [32]. Competence-stimulating peptide, a population density signaling molecule used by bacteria, can also activate mast cells through MRGPRX2, allowing mast cells to limit growth of bacteria that secrete this peptide [49].

Mast cell activation can facilitate the clearance of infections. Degranulation releases proteases and other antimicrobial molecules that have direct effects on pathogens [50,51,52,53]. Mast cells directly interact with bacteria via their adhesion receptors and capture pathogens in nets of extruded DNA called extracellular traps [54,55,56,57,58,59,60]. Critically, mast cells release many chemokines responsible for recruiting other immune cells. Neutrophils, recruited by TNF-α and mast cell protease 6, are essential for fighting bacterial infections [61,62], while IL-8 can recruit natural killer cells to fight viruses [63]. TNF-α is also used to recruit neutrophils and macrophages to form granulomas around cells infected with intracellular parasites [64]. Mast cells can also recruit monocytes through monocyte chemoattractant peptide-1 (MCP-1) [65]. Signaling molecules like histamine lead to vasodilation and vascular permeability, allowing influx of leukocytes from the blood into affected tissue [66]. Ultimately, a large amount of cytokines and chemokines are secreted after pathogen exposure and the exact functions of individual signaling molecules may vary between even similar pathogens and model systems [20,67]. Mast cells can also limit inflammation, mainly through the release of the anti-inflammatory IL-10 [68,69,70]. Given their role in immunity, it is not surprising that many potential therapies target mast cells to influence the outcome of infections (Figure 1).

### 2.1. Mast Cell Activators Used to Treat Infection

Mast cell activators can promote survival in several animal models of infection. Arifuzzaman et al. found that mastoparan, a small peptide derived from wasp venom that activates mast cells through the MRGPRX2 receptor, was able to enhance clearance of *Staphylococcus aureus* in a mouse model of skin infection [71]. Activation of mast cells after infection was able to drive neutrophil recruitment to the site of infection, enhancing clearance of bacteria from and healing of the infected skin lesion. This treatment is especially poised for use as a therapeutic, as it was topical and could avoid harmful systemic effects of mast cell activation. Furthermore, since it targets the immune system rather than the bacteria itself, it may prove to be more widely applicable and less subject to bacterial resistance than antibiotics. In a mouse model of malaria, injection of mice with an anti-IgE antibody or c48/80 was found to decrease parasite burden [72]. Mast cell secretion of TNF-α was thought to play an essential role in limiting the spread of the parasite in red blood cells. Mast cells can also be utilized to fight viruses, as pretreatment of mice with a TLR2 agonist protected from vaccinia virus infection [73]. Mast cell deficient mice reconstituted with mast cells from TLR2 knockout mice showed larger lesion sizes than wild-type mice, demonstrating the importance of mast cells protection from vaccinia virus infection. The ability to inhibit viral replication were also reduced when mast cells were deficient in cathelicidin [73], which is part of a group of antimicrobial peptides previously shown to be important for mast cell-mediated killing of bacteria [51].

Mast cell mediators can neutralize venoms and bacterial toxins, which is not surprising given their prominent role in protecting the host in these circumstances. Two studies have shown that antivenom IgEs significantly increased survival after injection of snake or bee venom into mice [74,75]. This effect was likely through release of mast cell proteases rather than direct neutralizing effects of the antibody, as activation of mast cells with a non-venom IgE/antigen combo improved survival to a similar extent as anti-venom IgEs [74]. Given mast cell proteases are known to cleave many venoms and toxins, drugs that can cause mast cells to degranulate may prove effective in limiting damage after venomous animal bites. A recombinant mast cell protease, tryptase, was found to reduce the toxicity of several different snake venoms in vitro using a zebrafish embryo model, even 30 min after snake venom exposure, suggesting purified tryptase may have therapeutic value as a general purpose post-exposure venom treatment [76].

### 2.2. Mast Cell Inhibitors Used to Treat Infection

Although mast cells play important roles in resolving bacterial and protozoan infections, several studies have found that inhibiting mast cells can have a positive impact on outcomes. In contrast to the Furuta et al. study mentioned above [72], Huang et al. found that treatment of a mouse model of malaria with c48/80 increased parasite burden and mortality [77]. Huang et al. also found that cromoglycate, which inhibits mast cell degranulation, improved survival and reduced parasite burden in a model of *Plasmodium berghei* infection [77] suggesting that mast cell activation exacerbated the infection. The different results obtained by Furuta and Huang may be due to differences in the severity of infection since Huang et al. used 10 times more infected red blood cells from donor mice to establish infection than Furuta [72,77]. The two studies also used different inbred mouse strains, which may have different responses to infection and mast cell activation. Another study by Huang et al. using *Toxoplasma gondii* in this same mouse model (in the KunMing mouse strain) also found that c48/80 increased parasite burden and inflammation while cromoglycate decreased disease [78]. C48/80 was found to modulate cytokine expression towards a Th1, rather than Th2, profile, which could explain increases in inflammation [78]. Interestingly, tryptase has been found to cleave many Th2 cytokines, offering a possible mechanism for skewing towards a Th1 profile [79]. Chiba et al. also found that c48/80 and cromoglycate had negative and positive effects, respectively, on recovery from *Chlamydia pneumoniae* lung infection [80]. In this particular model, it was found that activation of mast cells caused mast cell proteases to degrade tight junction proteins, facilitating bacterial and immune cell invasion [80]. In a *Trypanosoma cruzi* mouse model, though, cromoglycate increased parasite burden and mortality, showing that mast cell stabilization may be harmful under certain circumstances and in response to specific parasites [81].

Since mast cells populate the subepithelial lining of the lung and produce significant quantities of proinflammatory mediators such as IL-1, they play an important regulatory role in lung pathology during viral infections [82]. In mice infected with the highly pathogenic H5N1 influenza strain, infection was found to cause mast cell degranulation and mast cell stabilizers reduced lung damage and improved survival [83,84]. Here, mast cell stabilizers had no effect on virus levels, but instead prevented release of many proinflammatory mediators responsible for exacerbating damage to the lung epithelia [83,84]. Cromoglycate treatment has also been proposed as a treatment option to reduce inflammation in coronavirus disease 2019 (COVID-19), but would likely be most effective in patients with severe COVID-19 symptoms in the later stages of hypercytokinemia [85,86]. Certainly, dexamethasone is the only currently effective mediation for treating severe COVID-19, and it effectively inhibits mast cell proinflammatory mediator release similarly to later generation versions of this corticosteroid [87].

A few studies have explored the role of mast cells in vascular leakage in the context of infection. Tessier et al. found that cromoglycate could inhibit vascular leakage induced by *Bacillus anthracis* edema toxin [88]. Edema toxin acts rapidly, showing effects within 30 min of administration, but is not able to directly induce cell monolayer permeability nor degranulation of mast cells [88]. Instead, edema toxin requires host signaling molecules, including histamine, to exert its effects, though the mechanism is still unknown [88]. Similarly, cromoglycate was found to decrease intestinal permeability after infection with enterotoxigenic *Escherichia coli* K88 [89]. Like edema toxin, this *E. coli* strain requires mast cell mediators to induce permeability and inhibiting mast cells has a therapeutic effect [89]. Inhibition of mast cell degranulation with ketotifen also reduced visceral hypersensitivity in a rat model [90]. Interestingly, these last two studies found that specific strains of probiotic bacteria could have similar effects to degranulation inhibitors, suggesting the intestinal microbiome may influence disease partly through its interaction with mast cells [89,90]. Cromoglycate was also found to reduce intestinal permeability caused by early weaning in piglets [91], suggesting mast cell inhibitors may have more general therapeutic effects on diseases affecting intestinal permeability.

In models of sepsis, mast cells offer protection in moderate severity models, but can increase mortality in severe models [92,93]. Use of mast cell stabilizers after infection or depletion of mast cell granules before infection have both been found to improve survival [92,94,95,96]. In these severe models, mediators released from mast cells, like histamine, TNF-α and IL-1β, cause septicemic shock and prevent migration of neutrophils to the site of infection, which is essential for clearing bacterial infection [92,93]. Part of the issue is that mast cells seem to induce a systemic response rather than a local response when high levels of bacteria are present. This was shown when a lethal dose of bacteria in one location of a mouse prevented recruitment of neutrophils to a different location that received a sublethal dose [92]. Histamine seems to be a major factor in mortality, as histamine antagonists improve survival [95,96]. Interestingly, a mast cell deficient mouse model did not show improved survival in severe sepsis models and, unlike wild-type mice, showed no improvement in mortality upon treatment with cromoglycate [94]. This suggests that mast cells may play an important role in fighting sepsis, and only one aspect of their reaction, degranulation, may exacerbate disease. This reinforces a common theme in targeting mast cells for therapeutic purposes: mast cells can often both reduce and exacerbate disease and therapeutics must be chosen carefully to balance these activities rather than activating or inhibiting mast cells non-specifically. Notably, pretreatment of two different mouse strains, CBA and Swiss, with c48/80 had opposite effects on leukocyte recruitment [97]. On experimental induction of peritonitis, CBA mice pretreated with c48/80 showed increased recruitment of leukocytes, especially polymorphonuclear leukocytes like neutrophils, basophils and eosinophils. C48/80 pretreatment of Swiss mice, though, had the opposite effect, showing reduced influx of these cells in peritonitis. A possible explanation is that CBA mice have higher resting levels of mast cells in the peritoneum and recruit significantly more mast cells after pretreatment with c48/80 than Swiss mice [97]. Given these results, it is possible that patients may exhibit different responses to mast cell-targeting reagents in a clinical setting due to potential heterogeneity in human mast cell populations.

As illustrated above, only a few mast cell activators have been found to have positive effects on pathogen clearance, while degranulation inhibitors seem to enhance clearance or prevent damage in more model systems. It is possible that since degranulation produces such rapid effects, it may be effective only very early in infection. The sensitivity of this response, which is necessary for it to act immediately after infection when levels of PAMPs are low, could be harmful when pathogen levels are high and can trigger unnecessary, excessive and less localized responses. For this reason, cromoglycate and other degranulation inhibitors are promising potential therapeutics for several diseases, such as sepsis and intestinal permeability. Given that cromoglycate has a long history of clinical use and a good safety profile [19], it is the most promising for these applications. Proving that the mechanisms of action mentioned in the above studies will translate from animal models to humans is the largest hurdle that still exists for both mast cell degranulation inhibitors and activators. Unfortunately, degranulation independent roles of mast cells are often understudied in this context. The fact that most mast cell knockout studies show increased parasite burden, even while studies with mast cell stabilizers suggest a pathological role [36], shows that mast cells have an important role to play in limiting infection. It is possible that degranulation may still be necessary to slow infection initially or degranulation independent effects of mast cells may still be important. This is underscored by the fact that, in many of the studies discussed above, mast cells protected against infection when pathogen burdens were low and exacerbated disease when burdens were high. Ultimately, greater understanding of how mast cells affect innate immunity will be needed to unlock their full therapeutic potential.

## 3. Acquired Immunity

Although generally thought of as an innate immune cell, mast cells are also able to influence the adaptive immune response. TNF-α released from activated mast cells recruits dendritic cells to sites of infection, which then traffic to draining lymph nodes where antigen presentation takes place [98,99]. Both through release of mediators, like histamine or exosomes, and direct interaction, including transfer of antigens from mast cells to dendritic cells, mast cells are also able to modulate the phenotype of dendritic cells, controlling the nature of the downstream T-cell response [100,101,102,103,104]. Mast cells not only recruit antigen presenting cells to sites of infection but can themselves be induced to express MHC class II molecules and present antigens to T-cells, but it is unknown how common or relevant this is to mast cell biology [105,106]. Presentation of antigens by MHC class II molecules allows mast cells to stimulate T cell proliferation and activation [107]. Intriguingly, mast cells can also cross-present antigens via MHC class I molecules to CD8+ T-cells, activating them and inducing their proliferation [108]. Furthermore, they are also able to recruit T-cells, both by relaxing the surrounding vasculature and secreting chemokines like CCL5 [37,109,110]. Inversely, mast cells can also be activated or suppressed by T-cells [105,111]. Mast cells can also activate B-cells through cell surface expression of CD40L [107]. Mast cells can also control downstream T and B cell responses through release of cytokines and other signaling molecules [107]. After degranulation, mast cell-derived small particles containing TNF-α are released and can traffic to peripheral lymph nodes, where they likely influence the resulting adaptive immune response [112]. Surprisingly, mast cells also seem to be essential for maintaining immune tolerance, at least in the context of skin allografts [113]. While degranulation can break tolerance, de novo synthesis of several mediators, especially IL-10, can be immunosuppressive, while other mediators, like TGF-β and IL-2, are critical for the development of regulatory T-cells [114]. Regulatory T-cells can also recruit mast cells to maintain tolerance [114]. This ability to influence dendritic cells, T-cells and B-cells, when combined with their early role in infection, makes mast cells ideal targets for enhancing adaptive immune responses.

### Mast Cell Activators as Vaccine Adjuvants

Numerous studies have been done looking at mast cell activators as adjuvants for vaccines (Table 1). Adjuvants are chemicals or biological substances used to enhance the immunogenicity of a vaccine and are especially important in vaccines that do not use a live, replicating organism as a source of antigen [115]. While the mechanism of action of many adjuvants are unknown, they typically work by activating the innate immune system [115]. Most often, the studies below use cholera toxin (CT) as a control since both the B and A1 subunits are commonly used and effective adjuvants in laboratory experiments [116,117]. One of the most widely used mast cell activators, c48/80, improved generation of neutralizing antibodies over antigen alone and to a comparable level as CT [118,119,120,121,122,123,124,125]. Antigens included bacterial toxins, bacterial vaccine strains and viral surface antigens. Some studies even demonstrated improvements in survival or disease progression compared to antigen alone when challenged with a pathogen [120,124,125]. Notably, a few of these studies found that c48/80 induces production of immunoglobulin A (IgA), an antibody class associated with immunity at mucosal surfaces [118,120,124], suggesting c48/80 may be particularly effective at targeting infectious agents that enter through these surfaces. When using c48/80 with a *Streptococcus pneumoniae* vaccine strain, Zeng et al. found that, unlike CT, c48/80 did not induce significant amounts of IgA in saliva or nasal washes [125]. Despite this, using c48/80 as an adjuvant resulted in superior protection when mice were intranasally challenged with a pathogenic strain of *Streptococcus pneumoniae* [125]. At least two studies found little if any positive effect on using c48/80 as an adjuvant [126,127]. This may have been due to the model systems, as one study utilized a species of teleost fish that may have limited similarity to the more typical mouse models [127], while the other study used a mouse *Toxoplasma gondii* model that was previously shown to exhibit worse parasite burden when treated with c48/80 [78]. Instead, the same group found that mice exposed to the mast cell inhibitor cromoglycate during vaccination showed better survival and lower parasite burden after subsequent infection [128]. This paper, though, did not look at effects on antibody titers and, given that both vaccine and cromoglycate alone improved survival when administered before infection, may simply have unrelated additive effects [128]. Although one paper found that mast cell deficient mouse models produced blunted responses when c48/80 is used as an adjuvant [120], Schubert et al. found that the adjuvant effect of c48/80 was independent of mast cells [129], and attributed this difference to the use of the Kit mutant mouse model WBB6F1 *W/W^v^*. Overall, c48/80 is likely an effective adjuvant for some vaccines, but may prove ineffective in others. Whether or not it actually functions through mast cells, though, is still an open question. Although c48/80 activates MrgprB2 [130], the mouse orthologue of MrgprX2, it has also been found to activate other signaling pathways and cell types [131,132,133].

Several other mast cell activators, primarily consisting of basic peptides that likely target MRGPRX2, have been tested as adjuvants for vaccines (Table 1). Melittin, derived from honeybee venom, and mastoparan-7, derived from wasp venom, are able to increase antibody titers compared to antigen alone when delivered intranasally [138,139]. The antimicrobial peptide LL-37, when fused to an antigen, was found to increase production of antigen-specific IgG and IgA to a comparable degree as CT, and induced a Th17 skewed response [137]. The role of mast cells was not evaluated in this paper, but LL-37 is known to activate mast cells through MRGPRX2 [142]. Polymyxin B and colistin, two antibiotics derived from the bacteria *Bacillus polymyxia*, were found to be effective nasal adjuvants, increasing titers of antigen specific IgG and mucosal IgA compared to antigen alone [140]. Two analogues of these molecules that cannot induce degranulation of mast cells showed reduced titers of antigen-specific IgA but not plasma IgG, once again suggesting that mast cell activators may be particularly effective when targeting pathogens that enter at mucosal surfaces. When testing the adjuvant effect of several IL-1 family cytokines administered with recombinant influenza hemaglutanin (rHA), IL-18 was one of four cytokines that produced not only significant rHA specific IgG but also high amounts of rHA specific IgA on mucosal surfaces [136]. While the other cytokines were found to be mast cell independent, IL-18 showed much lower adjuvant activity in mast cell deficient mice. Notably, IL-18 provided the best protection from virus infection compared to the other IL-1 family cytokines, highlighting the potential of targeting mast cells to improve vaccine efficacy.

Several recent studies have shown the efficacy of chitosan nanoparticles to deliver antigens for vaccination. While nanoparticles are generally used to protect the antigen and release it at a target location within the cell, they are also capable of acting as adjuvants by increasing uptake of antigen by antigen presenting cells and inducing the inflammatory response [143]. Chitosan, a cationic polysaccharide derived from chitin found in crustaceans or fungi, offers several distinct advantages for use in a mucosal nanoparticle vaccine: low toxicity, low cost, adhesion to mucosal surfaces and immune activating properties [144]. When used to make a vaccine against *Bacillus anthracis* protective antigen, chitosan nanoparticles significantly increased the amount of antigen-specific IgGs [118]. When combined with c48/80, chitosan nanoparticles were particularly effective at increasing antigen specific IgA at mucosal surfaces. Another study by this same group also showed c48/80 loaded chitin nanoparticles could be used to increase the efficacy of a vaccine for Hepatitis B surface antigen, producing similar levels of antigen-specific serum IgG and mucosal IgA as an aluminum adjuvant [122]. Combining chitosan with poly-ε-caprolactone (PCL) in nanoparticles resulted in greater ability to trigger mast cell degranulation and antigen-specific IgG production, outperforming a commercial vaccine for Hepatitis B [141]. Overall, nanoparticles are a promising technology for use in vaccines, and combining them with mast cell-targeting adjuvants may prove particularly efficacious.

Though much interest has been dedicated recently to reagents that target mast cells, some evidence suggests that mast cells may play an unappreciated role in the function of existing adjuvants already used in some vaccines. A non-toxic cholera toxin A1 fusion protein, CTA1–DD, that binds to and forms immune complexes with IgGs was found to degranulate and stimulate TNF-α production from mast cells in vitro and in vivo [145]. Adjuvant activity of CTA1-DD was significantly lower in mast cell deficient mice, but only when delivered as an immune complex with IgG [145]. The cyclic lipopeptide surfactin also activates mast cells and displays lower adjuvant activity in mast cell deficient mice [146]. Many more possibilities exist that are unexplored, especially given that numerous vaccines target TLR receptors [115], most of which are present on mast cells. Despite these associations, it is important to note that the mast cell deficient mouse models used in these studies are controversial [129], and activation of mast cells does not necessarily imply that mast cells play a role in the humoral immune response [147].

Many mast cell activators have proven effective as adjuvants for various vaccines. Mast cell activators are most efficacious at inducing mucosal responses, with many of these activators producing superior titres of IgA antibodies at mucosal surfaces compared to other adjuvants. For this reason, they may prove most applicable against respiratory pathogens that utilize these surfaces as points of entry and replication. The potential safety of these adjuvants has been discussed, since they could potentially cause anaphylaxis [29]. With the exception of one study [123], most studies find little if any production of IgE when using mast cell activators as adjuvants [118,119,141]. Yoshino et al. also report no adverse effects on the olfactory bulb or kidney when using polymyxins as adjuvants [140]. An important side effect that may be missed in animal models is pain. Mast cells in general and MrgprX2 in particular have been implicated in pain, itch and neurogenic inflammation [32,148], which could limit their use in healthy subjects. Overall, given their safety and effectiveness, mast cell activators warrant more investigation as vaccine adjuvants but, like most potential mast cell targeting therapeutics discussed in this review, require a lot more study in clinical settings to determine whether results translate from animal models to humans.

## 4. Wound Healing and Angiogenesis

After formation of a wound, mast cells play an important role in the healing process. They are potentially involved in all three phases of wound healing: inflammation, proliferation and remodelling [149,150]. Since mast cells are present in the subepithelial layer of the skin, they are one of the first cells to respond to a wound. Furthermore, mast cells degranulate in response to neuropeptides that are released from injured sensory neurons [149]. Mast cell granule components, especially histamine, result in vasodilation and influx of leukocytes [150] while more mast cells are recruited the site of injury by the chemokine MCP-1 [151]. Several growth factors released by mast cells stimulate keratinocyte and fibroblast migration and proliferation, resulting in wound closure [152]. Mast cell proteases also play a role, as MCP9 degrades the basement membrane, allowing greater infiltration of fibroblasts, and tryptase activates factors that stimulate fibroblast proliferation [150,153,154]. Many of the mediators secreted by mast cells, such as VEGF and angiogenin, induce angiogenesis, helping to revascularize the wound [23,155]. Mast cells can induce the differentiation of fibroblasts into myofibroblasts, which contract to reduce wound area, and interaction between SCF on fibroblasts and c-kit on mast cells enhances wound contraction [156,157]. Pathological conditions surrounding wound healing may also involve mast cells. For example, mast cells are thought to contribute to scar formation and fibrosis and may play a role in persistence of ulcers [152,158,159]. Overall, the role of mast cells in wound healing is unclear since mast cell deficient models have shown inconsistent results [160,161,162,163,164]. Potential therapeutic approaches, therefore, may need to carefully consider the type and context of the wound they are modeling, as it is possible mast cells could play a lesser or opposite role in different circumstances.

Several drugs that inhibit mast cell degranulation have been tested for their ability to affect wound healing, mainly with regard to excessive scar formation. Mast cells, likely through their interactions with fibroblasts, seem to enhance scar formation, with mast cell deficient mouse embryos exhibiting smaller scar width after experimental wounding than their wildtype littermates [165]. Treatment of mice with cromoglycate after experimental wounding resulted in smaller scar width, reduced levels of proinflammatory cytokines and more normal appearing collagen fibrils compared to PBS treated animals [166]. Notably, resulting scar tissue was not any weaker and the rate of re-epithelialization was not reduced in cromoglycate treated mice [166]. Although cromoglycate reduced the amount of mast cells at the wound after 24 h post-wounding, it did not reduce the amount of tryptase β1. Since tryptase induces differentiation of fibroblasts into myofibroblasts, which should enhance scarring, this brings into question what the ultimate mechanism of cromoglycate is in this model [166]. Cromoglycate was also found to reduce inflammation and fibrosis around several different types of mesh implants in mice [167], leading further support to its applicability in moderating wound healing. In a pig model of hypercontractile scars, ketofin, an inhibitor of mast cell degranulation and histamine [19], reduced wound contraction and resulted in collagen organization more similar to unwounded skin [168]. In this model, ketofin not only reduced the amount of mast cells at the wound site, but also decreased the amount of myofibroblasts. Despite these positive effects on wound healing, inhibiting mast cell degranulation and the release of histamine could potentially have negative effects. For example, treatment of experimentally induced wounds with an antihistamine in a rat model resulted in decreased breaking strength, suggesting mast cells are important for maintaining the integrity of the wound closure at a critical time in wound healing [169]. Another study found that treatment with the histamine receptor antagonist famotidine resulted in poorer healing of anastomosis after colorectal surgery in rats [170]. Overall, inhibition of mast cell degranulation may be effective in reducing scar formation, but caution is warranted given that the full effects on the resulting wound are not yet fully understood.

Given that mast cells play an important part in initiating and progressing wound healing, it is not surprising that several studies have targeted their activation to enhance this process. For example, preoperative intraperitoneal injection of c48/80 improved healing of the mesenteric membranes after surgical incision [171]. Whether this was due to activation of mast cells or due to depletion of mast cell granules is unknown, as preoperative injection resulted in quicker closure of perforations than post-operative injection. In a diabetic rat model, treatment with naltrexone, an opioid receptor antagonist, was as effective as the standard of care for improving wound healing [172]. Notably, naltrexone increased both the amounts of mast cells at the healing site and the amount of angiogenesis [172,173]. While it is not known whether naltrexone affects mast cells directly, it is likely that an increase in their numbers would improve healing in diabetic wounds, which typically display lower amount of mast cells than non-diabetic models [172]. Mast cells can also be used more directly to promote healing. In a rat model, Karimi et al. induced ischemia by ligation and resection of several arteries and veins in the right leg [174]. After transection, the femoral artery was immersed in a solution containing mast cells derived from the bone marrow of mice and chitosan, a polysaccharide that has favourable effects on angiogenesis. This solution, in comparison to PBS or chitosan alone, resulted in greater angiogenesis as evidenced by an increase in blood vessels, especially those with a large diameter [174]. Although the parameters investigated were rather limited, the effects of mast cells on vascularization after ischemia warrants further study.

Compared to other applications, the number of studies targeting mast cells in wound healing is especially scant. Nevertheless, the studies that do exist suggest that mast cell-targeting therapies may be a promising approach. Like its other applications, mast cells appear to play a positive role in wound healing initially, but can have pathogenic consequences if activated excessively. For this reason, both inhibitors and activators could prove useful, the former for reducing scarring and fibrosis, and the latter for speeding healing and vascularization early on or closing chronic wounds. Ultimately, further work is required to prove the efficacy of targeting mast cells in wound healing and any clinical applications are unlikely in the near future.

## 5. Conclusions

We have studied the role of mast cells in homeostasis and disease for over a century and yet we still know so little about how they regulate their microenvironment. As our understanding of mast cells grows, greater attention is being paid to their roles in maintaining homeostasis. Mast cells regulate immune responses to infection, both as components of the innate immune system and through modulation of the adaptive immune system. They are also active modulators of the complex process of healing wounds. These roles make mast cells ideal targets of novel adjuvant design. However, the search for drugs that specifically target mast cells come with important caveats. For almost every biologic process in which mast cells are involved, mast cells demonstrate harmful effects when activated too aggressively or chronically. It is likely that the most important role of mast cells in any tissue is to maintain balance and bring disparate cellular signals back to a resting state. For this reason, activating or inhibiting mast cells must be done in the right temporospatial context.

Though many potential therapeutics targeting mast cells have been identified outside of their role in allergy, few of these strategies have made it as far as clinical testing. Ultimately, more work is needed to uncover the therapeutic potential of strategies targeting these cells. Ideally, drugs that target other aspects of mast cell functions, other than degranulation, must be explored, as mast cell release of lipid mediators, release of gases (such as nitric oxide), production of reactive oxygen species and production of growth factors, chemokines, cytokines and proteases is more likely to play a role in these circumstances. Additionally, strategies that more specifically target mast cells may help elucidate the exact roles they play in tissue homeostasis. Mast cells also present an additional complication in that they are phenotypically plastic—changing characteristics in both space (tissues) and time (with age). Heterogeneity exists not only between disease models and species, but also even within species and organisms. Future work will require careful attention to the disease models used, dosing of drug interventions and route of administration. With more research, future therapies may be able to tip the balance from the pathogenic to homeostatic functions exhibited by mast cells in the many roles they play in health.

## Figures and Tables

**Figure 1 cells-09-02713-f001:**
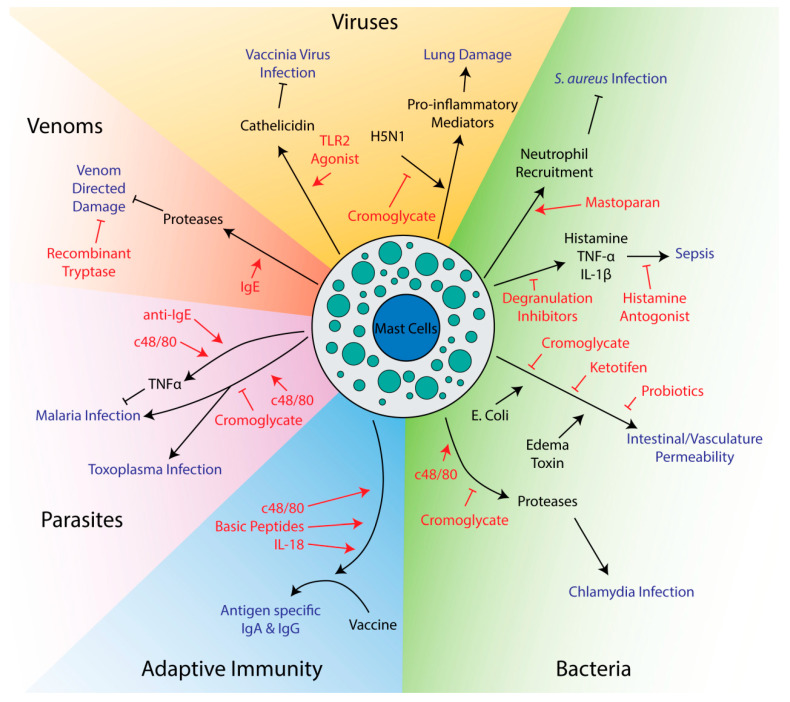
Potential therapeutics targeting mast cells in innate and adaptive immunity. Drugs and therapeutics are shown in red, while disease outcomes are shown in blue. Arrows indicate activation while flathead lines indicate inhibition.

**Table 1 cells-09-02713-t001:** Vaccine studies using adjuvants that target mast cells.

Adjuvant	Antigen	Host Species	Outcome	Details	Reference
C48/80	*Bacillus anthracis* protective antigen	Mice	Produced high levels of neutralizing antibodies, especially mucosal antibodies	Incorporated on chitosan nanoparticles.	[118,134]
C48/80	*Bacillus anthracis* protective antigen	Mice	Induced comparable levels of neutralizing antibodies compared to Cholera toxin and CpG adjuvants		[119]
C48/80	*Bacillus anthracis* protective antigen	Mice	Produced comparable levels of antibodies as cholera toxin adjuvant. Improved viability of macrophages exposed to anthrax toxin in vitro. Produced IgA at mucosal surfaces.		[120]
C48/80	*Bacillus anthracis* protective antigen	Rabbits	Greatly improved generation of neutralizing antibodies compared to antigen alone.	Found to be effective even when stored as a powdered formulation for two years.	[135]
C48/80	Hcβtre (botulinum neurotoxin A immunogen)	Rabbits	Increased the amounts of neutralizing antibody to a comparable degree as cholera toxin		[121]
C48/80	Hepatitis B surface antigen	Mice	Similar IgG induction to aluminum adjuvant.	Incorporated on chitosan nanoparticles. Generated TH2 and mucosal response.	[122]
C48/80	Ovalbumin	Mice	Generated more antigen-specific IgG and IgE	Antibody producing cells found to be localized to nasal surface where vaccine was administered.	[123]
C48/80	Recombinant H1N1 Influenza hemagglutinin protein	Mice	Improved IgG/IgA production and survival to a similar extent as cholera toxin adjuvant.		[124]
C48/80	*Streptococcus pneumoniae* vaccine strain SPY1	Mice	Improved IgG titers, but not IgA, compared to SPY1 alone, comparably to cholera toxin. Improved nasal clearance and survival, often to a slightly higher extent than cholera toxin.		[125]
C48/80	UV Attenuated *Toxoplasma gondii*	Mice	No improvement in survival, worse parasite burden than antigen alone		[126]
C48/80	Vaccinia virus B5R protein	Mice	Significantly improved survival compared to antigen alone.		[120]
C48/80 and Histamine	Heat killed *Vibrio anguillarum*	Gilthead seabream	Neither compounds resulted in a statistically significant increase in IgM B cells		[127]
Disodium Cromoglycate	UV Attenuated *Toxoplasma gondii*	Mice	Increases survival and reduces pathogen burden.	Effects on antibody titers not tested. Could be additive effects of vaccine and adjuvant separately, as both on their own improved survival.	[128]
IL-18 and IL-33	Recombinant influenza virus hemagglutinin	Mice	Induced antibodies and improved survivability, comparable to cholera toxin.	Many IL-1 family cytokines tested, but only IL-18 and IL-33 reduced in mast cell deficient mice.	[136]
LL-37	Dengue virus envelope protein domain III	Mice	Increased the amount of IgA and IgG antibody, comparable to cholera toxin.		[137]
LL-37	EGFP	Mice	Increased the amount of IgA and IgG antibody, comparable to cholera toxin.		[137]
Mastoparan-7	Cocaine	Mice	Improves generation of IgG and saliva IgA towards antigen and offered superior protection against cocaine induced locomotion.		[138]
Melittin	Tetanus toxoid	Mice	Induced higher levels than antigen alone		[139]
Melittin	Diphtheria toxoid	Mice	Induced higher levels than antigen alone		[139]
Polymyxin B and Colistin	ovalbumin	Mice	Increased amounts of antigen specific antibodies compared to antigen alone.		[140]
Poly-ε-caprolactone	Hepatitis B surface antigen	Mice	Increased antigen-specific IgG1 and IgG2c titers compared to antigen alone and generally to higher levels than a commercial vaccine.	Incorporated on chitosan nanoparticles.	[141]

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
