# Peer review of "Harnessing the Power of Mast Cells in unconventional Immunotherapy Strategies and Vaccine Adjuvants"

_cells, 2020, doi:10.3390/cells9122713_

Round 1

Reviewer 1 Report

In this comprehensive review entitled “Harnessing the power of mast cells in unconventional immunotherapy strategies and vaccine adjuvants”, the authors provided a summary of mast cell roles in innate and adaptive immunity as well as their role in wound healing and angiogenesis in vivo. However, some crucial information should be addressed.

Major comments

  1. Mast cell express a variety of pattern recognition receptors, including Toll-like receptors (TLRs). Mast cells respond to TLR ligands by secreting cytokines, chemokines, and lipid mediators, and some studies have found that TLR ligands can also cause degranulation. More details on mast cells and TLRs are needed.
  2.  
  3. Antimicrobial peptides function as important effectors of innate immunity. Authors should cover in detail the antimicrobial peptides known so far that affect mast cell function. (Adv Immunol 2017;136:123-162; J Dent Res. 2020 Jul;99(8):882-890)

  1. The authors’ main focus on this topic was the therapeutic interventions that show direct effect in vivo, the biological differences between animal models used in the studies and human should be addressed.
  2.  
  3. Although the authors provide a thorough list of studies and what have been found regarding mast cells and their involvement in innate and adaptive immunity, wound healing and angiogenesis, a proposed idea or speculation should be mentioned and discussed.
  4. MCs can modulate both the innate as well as adaptive immunity by regulating dendritic cells, B cells, natural killer (NK) and T cells by acting as antigen presenting cells (APCs). This is the most interesting aspect of mast cells, the antigen presenting property which is pivotal for effective vaccine response.. This has not been focused in this review. Moreover, MCs recruit and regulate the immune cells via immune synapses, exosomes to promote and facilitate antigen transfer and viral clearance. Activated MCs can secrete chemoattractant and recruit monocytes and neutrophils and promote differentiation of B cells into effector cells. Selective, critical and early MC responses to viral infections occur without a requirement for MC degranulation. These aspects have not been covered in the review.
  5. Authors may consider that readers may not have expertise on mast cells or adjuvants, as such at least some details should be provided on adjuvants and how they work.
  6. Following are some important literature on mast cells that may be considered.
  7.  

Lotfi-Emran, S., et al. Human mast cells present antigen to autologous CD4(+) T cells. J Allergy Clin Immunol 141, 311-321 e310 (2018).

Gaudenzio, N., et al. Cell-cell cooperation at the T helper cell/mast cell immunological synapse. Blood 114, 4979-4988 (2009).

Suto, H., et al. Mast cell-associated TNF promotes dendritic cell migration. J Immunol 176, 4102-4112 (2006).

Malaviya, R., Ikeda, T., Ross, E. & Abraham, S.N. Mast cell modulation of neutrophil influx and bacterial clearance at sites of infection through TNF-alpha. Nature 381, 77-80 (1996).

Nakae, S., et al. Mast cells enhance T cell activation: importance of mast cell costimulatory molecules and secreted TNF. J Immunol 176, 2238-2248 (2006).

Jawdat, D.M., Rowden, G. & Marshall, J.S. Mast cells have a pivotal role in TNF-independent lymph node hypertrophy and the mobilization of Langerhans cells in response to bacterial peptidoglycan. J Immunol 177, 1755-1762 (2006).

Stelekati, E., et al. Mast cell-mediated antigen presentation regulates CD8+ T cell effector functions. Immunity 31, 665-676 (2009).

Mekori, Y.A., Hershko, A.Y., Frossi, B., Mion, F. & Pucillo, C.E. Integrating innate and adaptive immune cells: Mast cells as crossroads between regulatory and effector B and T cells. Eur J Pharmacol 778, 84-89 (2016).

Shelburne, C.P., et al. Mast cells augment adaptive immunity by orchestrating dendritic cell trafficking through infected tissues. Cell Host Microbe 6, 331-342 (2009).

St John, A.L., et al. Immune surveillance by mast cells during dengue infection promotes natural killer (NK) and NKT-cell recruitment and viral clearance. Proc Natl Acad Sci U S A 108, 9190-9195 (2011).

Mantri, C.K. & St John, A.L. Immune synapses between mast cells and gammadelta T cells limit viral infection. J Clin Invest 129, 1094-1108 (2019).

Galli, S.J. & Gaudenzio, N. Human mast cells as antigen-presenting cells: When is this role important in vivo? J Allergy Clin Immunol 141, 92-93 (2018).

Burke, S.M., et al. Human mast cell activation with virus-associated stimuli leads to the selective chemotaxis of natural killer cells by a CXCL8-dependent mechanism. Blood 111, 5467-5476 (2008).

Carroll-Portillo, A., et al. Mast cells and dendritic cells form synapses that facilitate antigen transfer for T cell activation. J Cell Biol 210, 851-864 (2015).

Frossi, B., Mion, F. & Pucillo, C. Deciphering new mechanisms on T-cell costimulation by human mast cells. Eur J Immunol 46, 1105-1108 (2016).

Skokos, D., et al. Mast cell-dependent B and T lymphocyte activation is mediated by the secretion of immunologically active exosomes. J Immunol 166, 868-876 (2001).

Choi, H.W., et al. Perivascular dendritic cells elicit anaphylaxis by relaying allergens to mast cells via microvesicles. Science 362(2018).

Palm, A.K., Garcia-Faroldi, G., Lundberg, M., Pejler, G. & Kleinau, S. Activated mast cells promote differentiation of B cells into effector cells. Sci Rep 6, 20531 (2016).

Merluzzi, S., et al. Mast cells enhance proliferation of B lymphocytes and drive their differentiation toward IgA-secreting plasma cells. Blood 115, 2810-2817 (2010).

Minor comments

  1. Please carefully check the typographical and spacing errors. There should be a space before the citation/bracket (i.e. line 109, 227, etc.).
  2. Line 111: the authors mentioned that 4 key areas that will be discussed, however, they only specified 3 of them.
  3. Line 434: change ‘effects’ to ‘affects’

Reviewer 2 Report

The manuscript by Willows and Kulca is a comprehensive and well written review of the different studies of the role of mast cells and mast cell products in immunity and disease.

The aim is apparently to identify novel avenues to treatment of mast cell related diseases, however, the major strength in my mind with the review is the summary of the extremely disparate studies and results coming from studies of mast cell and their role in different diseases. The new strategies put forward are actually the weakness of the review, very few convincing such strategies are presented not even in the vaccine adjuvant section where the results are relatively discouraging for any kind of commercial strategy as the compounds tested not only act on mast cells but numerous other cells and thereby are difficult to control.

Minor comments:

Line 54: mast cells do not only originate from the bone marrow. The absolute majority does similar to most macrophages originate from an early wave of colonization from the yolk sac and the bone marrow seems mainly to supply mast ell precursors and to repopulate during inflammatory conditions.

Line 84 and 85: the statement of TLR s om mast cells is primarily based on studies of very immature mast cell like cells in the form of BMMCs and not on mature tissue mast cells. Recent transcriptome studies also indicate very low transcript levels of TLR in mature mast cells and studies of LPS stimulation of peritoneal mast cells have also been negative indicating a minor role of TLRs in mast cell activation in vivo. This shows similar to many of the other effects by mast cells described in this manuscript that there are highly contradicting results coming from different labs that at least partly may come from using suboptimal cell or cell line models.

Line 196 197:  the role of C48/80 in modulate cytokine expression towards a TH1 phenotype may actually be depending an effect by mast cell chymase and tryptase in cleaving and thereby selectively inactivate key TH2 cytokines.

Reviewer 3 Report

The manuscript „Harnessing the power of mast cells in unconventional immunotherapy strategies and vaccine adjuvants” by Steven Willows and Marianna Kulka addresses the question how the modulation of MC activity affects innate and adaptive immunity reactions, and wound healing.

In the beginning, my expectations of this review were quite high. Unfortunately, when reading the manuscript, my interest declined. The authors’ promise to explain “…how certain strategies that specifically target and activate mast cells can be used to treat and resolve infections, augment vaccines and heal wounds“  was not fulfilled.

The reasons for this are as follows:

1) The effector role of mast cells in allergic reactions is so impressive and particularly anaphylactic reactions so overwhelming, that regulation of homeostasis is sometimes difficult to notice. In my view, the protective role of mast cells was shown for the first time by Echtenacher et al.  1996, doi: 10.1038/381075a0. The link between allergies and cancer to introduce the role of MCs in the maintenance of homeostasis is somehow misleading.

2) In the introduction, some errors need correction. The hypothesis explaining the origin of allergies, suggested by Margaret Profet (so called toxin hypothesis) is not based on observations that allergies negatively correlate with cancer, instead, it is based on common properties between allergies and response against toxins. The corresponding paper was published in 1991 (Profet M.  1991 doi: 10.1086/417049) – please, correct.

3) I would recommend to include and discuss exciting recent papers addressing dual MC development  (Gentek et al 2018; doi: 10.1016/j.immuni.2018.04.025, Li et al 2018, doi:10.1016/j.immuni.2018.09.023) and the development of skin MC populations (Weitzmann et al 2020 doi: 10.1016/j.jid.2020.03.963). Departing from those papers, the question arises whether the different developmental origin of MCs could lead to attenuated sensitivity to different stimuli. In line with this hypothesis, there is the observation regarding fetal sensitization by maternal IgEs, enhancing MC sensitivity to respond to allergens in early life. Corresponding findings have been published recently (Msallam et al 2020, doi:  10.1126/science.aba0864).

4) Line 103 – …many potential treatments which targets these cells are being developed. To my knowledge, there are not so many effective therapeutic options available – please, clarify.

5) An exciting observation is that MCs through the degranulation are indeed inactivating potentially harmful toxins. Here, I suggest starting with a discussion of initial publications and to include and discuss the role of MCs in the inactivation of endogenous toxins as well (Metz et al 2006 doi: 10.1126/science.1128877; Schneider et al 2007 doi:10.1084/jem.20071262).

6) Line 264 – in line with the observation that MCs could limit the infections, is an observation regarding MC activation by bacterial quorum-sensing peptides (Pundir et al 2019 doi:10.1016/j.chom.2019.06.003). Please, discuss and include.

7) Lines 84 and 366 - please, include a comment on the discrepancy regarding the TLR expression on MCs (Plum et al. 2020 doi: 10.1016/j.immuni.2020.01.012).

8) In my opinion, the Comp48/80 has a limited future as an adjuvant because the activation of MrgprX2 and Mrgprb2 could be accompanied by strong pain. Could you discuss this aspect as well?

9) An important aspect is the maintenance of peripheral tolerance. MC activation could not only enhance adaptive immune response, but also lead to a loss of tolerance. Please, discuss this aspect of MC activation, and relevance in e.g. transplantation experiments.

10) line 460 – “MC most important role of mast cells in any tissue is to maintain balance and bring disparate cellular signals back to a resting state”. How, in this context, can you explain the absence of any side effects in human skin after MC depletion by imatinib (11)?

Taken together, MC research should summarize and discuss sometimes contradictory findings. In my view, finding out how MC affects e.g. T- or B-cell response would be much straighter, if we looked upon the quality of T- and B-cell response – in particular organ- and pathogen-specific settings. Therapeutic applications might be suggested, if we understand, how different MC-derived signals are affecting local and systemic reactions. This could still take some time. We should look ahead confidently for single cell-based techniques, improved in vitro models, TLR-specific reporter cells/mice strains and conditional MC-specific genetic models to learn more about this fascinating cell type, to answer open questions and use this powerful cell type to modulate immune reactions.

Minor comments:

  1. - Lines 110 -111 – “…this review will instead focus on four other key areas… “– only three are mentioned – please, correct.
  2. - line 14 – “…mobilizing and amplifying the innate and adaptive immune system” – my suggestion would be” …mobilizing and amplifying the reactions of the innate and adaptive immune system”.

Round 2

Reviewer 1 Report

none

Reviewer 3 Report

I would like to thank the authors for the improvements.